# The Proteasome Lid Triggers COP9 Signalosome Activity during the Transition of *Saccharomyces cerevisiae* Cells into Quiescence

**DOI:** 10.3390/biom9090449

**Published:** 2019-09-04

**Authors:** Laylan Bramasole, Abhishek Sinha, Dana Harshuk, Angela Cirigliano, Gurevich Sylvia, Zanlin Yu, Rinat Lift Carmeli, Michael H. Glickman, Teresa Rinaldi, Elah Pick

**Affiliations:** 1Department of Biology and Environment, Faculty of Natural Sciences, University of Haifa at Oranim, Tivon 36006, Israel; 2Department of Human Biology, Faculty of Natural Sciences, University of Haifa, Haifa 31905, Israel; 3Department of Biology and Biotechnology "Charles Darwin", La Sapienza University of Rome, 00185 Rome, Italy; 4Department of Biology, Technion-Israel Institute of Technology, Haifa 32000, Israel

**Keywords:** 26S proteasome, proteasome lid, Rpn11, Cdc53, Cullin, SCF (Skp, Cullin, F-box containing complex), NEDD8 (neural precursor cell expressed developmentally down-regulated 8), Rub1 (Related ubiquitin 1), CSN (COP9 signalosome), *Saccharomyces cerevisiae*, diauxic shift, budding yeast

## Abstract

The class of Cullin–RING E3 ligases (CRLs) selectively ubiquitinate a large portion of proteins targeted for proteolysis by the 26S proteasome. Before degradation, ubiquitin molecules are removed from their conjugated proteins by deubiquitinating enzymes, a handful of which are associated with the proteasome. The CRL activity is triggered by modification of the Cullin subunit with the ubiquitin-like protein, NEDD8 (also known as Rub1 in *Saccharomyces cerevisiae*). Cullin modification is then reversed by hydrolytic action of the COP9 signalosome (CSN). As the NEDD8–Rub1 catalytic cycle is not essential for the viability of *S. cerevisiae*, this organism is a useful model system to study the alteration of Rub1–CRL conjugation patterns. In this study, we describe two distinct mutants of Rpn11, a proteasome-associated deubiquitinating enzyme, both of which exhibit a biochemical phenotype characterized by high accumulation of Rub1-modified Cdc53–Cullin1 (yCul1) upon entry into quiescence in *S. cerevisiae*. Further characterization revealed proteasome 19S-lid-associated deubiquitination activity that authorizes the hydrolysis of Rub1 from yCul1 by the CSN complex. Thus, our results suggest a negative feedback mechanism via proteasome capacity on upstream ubiquitinating enzymes.

## 1. Introduction

One of the major mechanisms to regulate proteostasis in cells is the posttranslational modifications of proteins by polypeptide modifiers, such as ubiquitin (Ub), and family members known as ubiquitin-like (Ubl) proteins [1]. Modification of proteins by Ubls is a process involved in various cellular responses, the most famous of which is protein modification by Ub and its comprehensive role in the Ub proteasome system (UPS) [2]. Ubiquitination is an ATP-consuming process that occurs through covalent attachment between a specific Lys residue in target proteins and Gly–Gly residues at the carboxyl terminus of Ub [3]. Ubiquitination requires a cascade of enzymes, which activate (E1), conjugate (E2), and eventually ligate (E3) ubiquitin to target proteins, resulting primarily in their proteasomal degradation [3,4]. Substrate selectivity of Ub attachment is determined by E3 ligases; each of them binds to specific substrate(s) and catalyzes their covalent attachment to Ub chains [5]. The modular family of Cullin-RING Ub ligases (CRLs) constitute the largest class of E3 ligases [6,7], represented by the archetypical Skp, Cullin, F-box containing complex (SCF) [5,8]. The SCF includes a Cdc53–Cullin1 (herein yCul1) scaffold subunit that interacts with the RING E3 subunit, Rbx1, via its carboxyl terminus and with the Cullin-specific adaptor protein (Skp1) via its amino terminus. Skp1 binds an F-box protein (Fbp) that serves as a substrate receptor, providing specificity to the enzymatic complex by selecting substrates for ubiquitination [5,8]. Cdc4 is an essential and highly characterized Fbp in *S. cerevisiae*. Cdc4 recognizes Sic1 for ubiquitination, leading to its proteasomal degradation [9]. Sic1 is an inhibitor of cyclin-dependent kinase (CDK)–cyclin B activity, and its degradation promotes DNA replication [9,10]. Sic1 is phosphorylated in the late G1 phase by G1-cyclin CDK (Cln-Cdc28) activity, which results in Cdc4 recognizing it [11]. Sic1 phosphorylation is antagonized by the cell cycle phosphatase, Cdc14, which leads to its stabilization, accumulation, and consequently to cell cycle arrest. Additional substrates of Cdc4 include the Cln–Cdc28 inhibitor Far1, the replication protein Cdc6, and the transcription factor Gcn4 [12]. Notably, Fbps are short-lived proteins that are switched off by autoubiquitination within their own SCF complex [13]. With dozens of Fbps identified in the human genome and hundreds of potential substrate adaptors to other Cullins, CRLs form hundreds of assemblages, modifying thousands of proteins by Ub, accounting for approximately one-fifth of all proteasome substrates [14,15].

CRLs are activated upon covalent modification of their Cullin subunit by NEDD8 (also known as Rub1 in *S. cerevisiae* and plants), the closest paralog of Ub. NEDD8–Rub1 is required for vitality in all studied organisms, with the notable exception of *S. cerevisiae* [16]. *S. cerevisiae* Rub1 is activated by a particular heterodimeric E1 enzymatic complex Ula1–Uba3, and transferred to its cognate E2 enzyme, Ubc12 [17]. In the next step, Ubc12–Rub1 interacts with Rbx1, triggering the transfer of Rub1 to a specific Lys residue on the Cullin [18]. Cullins undergo cycles of Rub1 conjugation and deconjugation (also known as NEDDylation and deNEDDylation) during their catalytic cycle. The deNEDDylation of Rub1 from Cullins is performed by the COP9 signalosome (CSN), an evolutionarily conserved multi-subunit complex [19]. In addition to deNEDDylation activity, the CSN also hinders both substrate recognition and ubiquitination by CRLs [20,21,22,23,24]. Although neither CSN nor Rub1 are strictly essential in budding yeast, CSN does drive canonical Cullin deNEDDylation in this model organism as well. However, its mechanistic purpose remains unclear [25,26,27,28]. With only six subunits, *S. cerevisiae* harbors the smallest and most diverged CSN complex, four of which contain a proteasome, CSN, and eIF3 (PCI) homology domain [29], one catalytic MPN+ subunit, Csn5–Rri1, and an endemic subunit harboring the S6CD domain [26,30,31,32]. As in all other model organisms, mutants in any *S. cerevisiae* CSN subunit share a characteristic biochemical phenotype of accumulated yCul1^R^ [19,26,30]. An interesting characteristic of CSN is the close homology to the 19S-lid, the distal part of the regulatory particle (RP) of the 26S proteasome [29,33], so much so that one of the subunits of *S. cerevisiae*, Rpn5, was identified as a bona fide subunit of both complexes [30,31,32]. Both, the CSN and the 19S-lid belong to the PCI family of complexes. Six subunits of canonical complexes bear the hallmark PCI domain alongside two subunits carrying an Mpr1–PAD1–N-terminal (MPN) domain [34]. In each complex, one of the MPN subunits is a catalytically active MPN+–JAMM (JAB1/MPN/Mov34 metalloenzyme) metalloprotease, namely Csn5 in CSN [25,35] and Rpn11 in 19S-lid [36,37,38]. In accordance, the substrates of these two enzymes are also close paralogues—Cullin–Rub1 for Csn5, and Ub conjugates for Rpn11 [39]. Single-site mutations in MPN+–JAMM residues of either Csn5 or Rpn11 cause a decrease in CSN deNEDDylation and proteasome deubiquitination (DUB), respectively [25,36,37]. In this study, we observed that certain mutants of Rpn11 in *S. cerevisiae* are characterized by accumulation of yCul1^R^ during the post-diauxic shift state prior to saturation. At this cellular phase, nutrients become limiting and yeast cells stop dividing [40]. Further characterization revealed that proteasome 19S-lid-associated DUB activity attributed to Rpn11 is essential before the hydrolysis of Rub1 from yCul1^R^ conjugates by the CSN complex. 

## 2. Materials and Methods

### 2.1. Yeast Strains and Growth Conditions

This study included widely used W303 and BY4741 laboratory strains of *S. cerevisiae*. The experiments obtained a similar pattern of results for both studied strains. Double mutants used in this study were generated by mating haploid strains, sporulation, and tetrad dissection. Genotypes and mating types of haploid progeny were determined by phenotype, auxotrophic analysis, polymerase chain reaction (PCR), and immunoblotting. Unless otherwise specified, *S. cerevisiae* cells were grown in standard growth conditions at a permissive temperature of 28 °C. All yeast strains were grown on glucose-rich YPD (yeast extract 1%, peptone 1%, dextrose 2%) medium (or in non-fermentable glycerol-containing YPG (yeast extract 1%, peptone 1%, galactose 2%) medium. Both YPD and YPG were complemented with adenine hemisulfate (0.004%). For all treatments, unless stated otherwise, starter cultures were grown overnight, diluted to OD_600_ = 0.5, and incubated for an indicated number of times, temperatures, or treatments, as described in the figure legends. Growth phases were determined according to Bramasole et al. 2019 [41] as follows: early logarithmic phase, 4–6 h; logarithmic phase, 6–8 h; diauxic shift, 10–12 h; post diauxic phase, >22 h. Plasmids were maintained by culturing plasmid-containing strains in a selective synthetic complete (SC) medium based on yeast nitrogen base (YNB) supplemented with ammonium sulfate, in which a complete mixture of amino acids supplements each of the commonly encountered auxothropies. For proteasome inhibition, MG132 (dissolved in dimethyl sulfoxide (DMSO)) was added for 2 h to either the YPD growth medium of the *Δpdr5* mutant strain or to a unique growth medium previously described for other strains [42]. Yeast strains and plasmids used in this study are listed in Table 1 and Table 2.

### 2.2. Vitality Test

To evaluate the viability of various single deletion mutants, a drop dilution assay in YPD was utilized. Overnight grown cultures in YPD were harvested and washed twice with sterile distilled water. Equal numbers of cells (confirmed by cell counting) were used for a ten-fold serial dilution, followed by spotting 2 µL of cultures on YPD plates that were then incubated at 28 °C for 24–48 h. For the double mutants, two hundred cells of each yeast strain at the logarithmic phase were plated in YPD in triplicate. Agar plates were incubated at 28 °C for 2–3 days. Grown colonies were counted. Vitality was estimated by calculating the percentage of viable cells out of the total number of cells that were initially plated. Each experiment was repeated three times.

### 2.3. Cell Harvest and Immunoblotting

To denature total cell extracts, cells were harvested in trichloroacetic acid (TCA) as previously described [30]. Samples were resolved by SDS-PAGE (sodium dodecyl sulfate - polyacrylamide gel) and transferred to a nitrocellulose membrane for immunoblotting. Experiments were repeated at least three times and a representative result is shown.

### 2.4. Calmodulin-Based Affinity Purification

Yeast cultures were grown overnight in SC medium and transferred for five additional hours at 34 °C. Cells were pelleted, washed twice with double-distilled water (DDW), and then washed once with chilled (4 °C) Calmodulin binding buffer (25 mM Tris [pH 7.4], 200 mM NaCl, 2 mM CaCl_2_, 0.2% NP-40, 5 mM NaF) complemented by β-mercaptoethanol and a cocktail of free-ethylene diamine tetra-acetic acid (EDTA) protease inhibitors.. The pellet was re-suspended in two volumes of the same buffer and lysed by glass beads at 4 °C. Clarified lysates were incubated with calmodulin sepharose beads (GE Healthcare) for 3–16 h at 4 °C and washed with 20 volumes of binding buffer. Bound proteins were eluted in elution buffer (the same as the binding buffer, but containing 2 mM ethylene glycol tetra-acetic acid (EGTA) instead of the 2 mM CaCl_2_). Eluted proteins were subjected to immunoblotting.

### 2.5. Histidine-Based Affinity Purification

Yeast cells ectopically expressing RGS–8His–yCul1 were grown in SC medium for 16 h at a permissive temperature of 28 °C. Cells were harvested and the pellet was washed twice with double distilled water; 250 OD_600_ of each strain were dissolved in chilled cell lytic Y cell lysis reagent (Sigma-Aldrich, St. Louis, Missouri; USA) complemented with EDTA-free protease inhibitor (Thermo Fisher Scientific, Waltham, MA USA) and extracted at 4 °C by glass beads. Clarified lysates were mixed in a 1:1 ratio with 2 × Ni–NTA binding buffer (50 mM Tris pH 7.4, 300 mM NaCl, 40 mM imidazole). For pulldown, Ni–NTA beads were added to 1mg of each lysate and incubated overnight at 4 °C. Beads were washed five times with 1 × Ni–NTA binding buffer and elution was performed with the same buffer complemented with 250 mM imidazole. Eluted proteins were subjected to immunoblotting.

### 2.6. Microscopy

#### 2.6.1. Light Microscope Imaging

To visualize cell morphology and cell cycle arrest, yeast strains were grown at 28 °C for 16 h and shifted to a restrictive temperature of 37 °C for 5 h to enhance cell cycle defects. Samples from the growing cells were taken directly to slides and visualized under light microscope with ×400 magnification.

#### 2.6.2. Confocal Imaging for Rpn5

For Rpn5 localization, W303 WT (wild type) and *rpn11–m1* with genomic tagged Rpn5–GFP at the precise chromosomal location were used. For insoluble protein deposit (IPOD) and juxta nuclear quality control compartment (JUNQ) localization, the aforementioned strains were further ectopically transformed with plasmids expressing either mCherry–VHL (von Hippel-Lindau) or mCherry–Rnq1. For live imaging, strains were incubated overnight at 28 °C in SC Ura medium, and later shifted from SC Ura medium containing 2% galactose to an initial 0.2 OD_600_ at 28 °C or 34 °C for 5 h. Aliquots of cells were immediately transferred to slides for live cell microscopy at room temperature under a confocal laser scanning microscope (Carl Zeiss, LSM 710, Oberkochen, Germany). Images were captured at ×63 objectives with a zoom of ×6.5; 0.5 μm interval Z-stack images were used for three-dimensional (3D) reconstruction using Canvas 10 software.

#### 2.6.3. Florescent Imaging

Cells were grown in YPD supplemented with adenine at 34 °C for 5 h. Aliquots of cells were immediately fixed and 4’,6-diamidino-2-phenylindole (DAPI) was added until a final concentration of 2.5 µg/mL was reached. Cells were visualized with fluorescence microscopy. The fluorescence was observed with filter sets (365 nm excitation and 445/450 nm emission) for DAPI. The microscope used was a Zeiss Axio Imager Z1 florescence microscope with an AxioVision 4.8 digital image processing system, and the objective lens was ×63 oil LSM 710 (Carl Zeiss).

### 2.7. Antibodies

The following antibodies were used: anti-Cdc53 (yCul1), anti-Sic1, anti-Cdc4, anti-Actin, and anti-CBP (Dako, Santa Cruz, CA, USA); anti-Ub (Dako); anti-Rpn11 and anti-Rpn12 [30]; anti-Rpn8 [43]; and anti-Rpn1 and anti-Rpn2 [44].

### 2.8. Inhibition of COP9 signalosome (CSN) Activity

The CSN5i-3 inhibitor [45] was provided by Novartis (Novartis Institutes for BioMedical Research, Basel, Switzerland) for Medical Research under the terms of the Novartis Transfer Agreement for academic research proposes. Cultures were grown overnight and diluted in YPD to 0.5 OD_600_ and grown for 6 h before the addition of CSN5i-3 from a 20 mM stock diluted in DMSO. Cullin NEDDylation status was assessed by immunoblotting. The ratio of yCul1^R^ to yCul1 in CSN5i-3-treated and untreated cells was quantified by IMAGEJ v1.40f software (http://imagej.net/). The inhibition was calculated as a ratio of the accumulated yCul1^R^ in treated and untreated cells. The average of three independent experiments was calculated, including standard deviation. 

## 3. Results and Discussion

### 3.1. Distinct rpn11 Mutants Accumulate yCul1^R^

Two distinct mutants of Rpn11 were employed in this study: the catalytic dead mutant *rpn11^D122/A^*, and *rpn11-m1*, a carboxyl-terminal truncated mutant lacking 31 amino acids, which were replaced by nine other amino acids due to a frameshift (mpr1-1) [37,53] (Figure 1A). Both *rpn11^D122/A^* and *rpn11-m1* are temperature-sensitive mutants that share physiological and biochemical phenotypes, such as cell cycle defects, accumulation of Ub chains, and defects in proteolysis [37,47]. However, *rpn11-m1* exhibits additional defects in 26S proteasome integrity by forming an incomplete 19S-lid subcomplex attached to 19S base-CP [43]; *rpn11-m1* also displays mitochondrial phenotypes when grown above a semi-restrictive temperatures (>32 °C) [41,54].

In a previous study we described a phenotype of rpn11-m1 that exhibits a high ratio of modified to unmodified yCul1 at the end of the logarithmic phase in 34 °C, distinguishing this mutant from rpn11^D122/A^ [41] (Figure 1B; 8 h). Further characterization revealed that *rpn11-m1* mutant cells lack a redox-dependent thiol switch-off of Cullin NEDDylation enzymes because of a failure to shift into mitochondrial respiration when glucose in the growth medium is exhausted (diauxic shift) [41] (Figure 1B; 8 h). Nevertheless, rpn11-m1 retained elevated levels of yCul1^R^ during the post-diauxic phase, a state in which constituent cells proliferate very slowly and readjust their metabolism to utilize other carbon sources present in the medium [40]. Supporting the notion that this case was not mitochondria-related, yCul1^R^ was also accumulated in the *rpn11^D122A^* mutant (Figure 1B; 24 h). High yCul1^R^ status was detected in both rpn11 mutants, even at a permissive temperature of 28 °C, at which respiration defects of *rpn11-m1* were diminished and Ubc12~Rub1 thioester forms vanished (Figure 1C,D; Appendix A) [41]. Respectively, high yCul1^R^ were detected when shifting overnight cultures to nutrient-rich growth media, including the non-fermentable carbon source glycerol (forcing mitochondrial respiration) as a sole energy source (Figure 1C). The high ratio of yCul1^R^ in rpn11 mutants correlating with accumulation of the short-lived proteasome substrate, Cdc4, due to defects in proteolysis [47,55] (Figure 1C, middle; Appendix A). The high ratio of yCul1^R^ is also observed in various rpn11-metalloprotease deficient MPN+–JAMM point mutants (*rpn11^H111A^*, *rpn11^S119A^*, and *rpn11^D122A^*), but not in silent mutations (*rpn11^C116A^* and *rpn11^D116S^*) (Figure 1A,D; Appendix A). Notably, the accumulation of yCul1^R^ differentiates rpn11 mutants from deletion mutants of other please define. None of the latter mutants exhibited this biochemical phenotype, neither when glucose in the growth medium was exhausted at the post diauxic phase or when the carbon source was replaced by glycerol (Appendix A).

### 3.2. The Accumulation of yCul1^R^ in rpn11 Mutants Is Not Associated with Cell Cycle Defects

The accumulation of yCul1^R^ could be associated with physiological or biochemical phenotypes shared between *rpn11-m1* and *rpn11^D122A^*. However, this phenotype may be an indirect consequence of their common cell cycle arrest phenotype [37,47] rather than a direct consequence of Rpn11 function (as a proteasome-associated DUB). Both *rpn11* mutants demonstrated morphology of elongated bud and pre-anaphase arrest. Previous studies showed that the overexpression of *CDC14*, a phosphatase of the mitotic exit, suppresses the associated cell cycle phenotypes in *rpn11-m1*, suggesting a positive genetic interaction between these proteins and location of Rpn11 upstream to Cdc14 [46]. However, although the overexpression of CDC14 restored defects in cell cycle it did not alter the accumulated yCul1R in rpn11-m1 (Figure 2B). In agreement, the cell cycle phenotype observed in *rpn11* mutants is shared with *cdc14-3* mutation of *CDC14* (Figure 2A). However, unlike *rpn11* mutants, the yCul1^R^ to yCul1 ratio between in *cdc14-3* is similar to WT (Figure 2C; top). The accumulation of yCul1^R^ in *rpn11* mutants also correlated with the accumulation of the SCF^Cdc4^ product Sic1 (Figure 2B; bottom). Remarkably, Sic1 is moderately accumulated in *rpn11^D122/A^* and highly accumulated in *rpn11-m1*. A possible reason for this discrepancy between the studied mutants is the essentiality of the carboxyl terminal domain of Rpn11 (missing only in *rpn11-m1* but not in *rpn11^D122/A^*) for the regulation of *CDC14* [46], which in turn is responsible for the turnover of Sic1 [56]. The above results suggest that the accumulation of *yCul1^R^* in *rpn11* mutants is not influenced by the cell cycle. This assumption is further supported by the finding that overexpression of *RPN8* in *rpn11-m1* partially suppresses defects associated with proteasome structure and cell cycle [47], but does not lead to alterations in yCul1 modification status (Appendix A). Likewise, NEDDylation status of yCul1 was not altered even upon synchronization of WT cells in the G1–S-phase boundary by α-factor-mediated arrest for 150 min followed by release back into the cell cycle (Figure 2D). Altogether, our findings suggest that the NEDDylation status of yCul1 is influenced by biochemical properties rather than by the cell cycle. 

### 3.3. Rpn5 Availability Is Sufficient in rpn11-m1

In *S. cerevisiae*, Rpn5 plays a dual role serving as both a 19S-lid and as a CSN subunit [30,32]. Before quiescent, Rpn5 together with other subunits of the 26S proteasome migrate into cytoplasmic structures named proteasome storage granules (PSG) [57]. PSGs protect proteasomes from rapid degradation via autophagy or from aggregation [58,59]. Proteasomes of *rpn11-m1* cannot migrate to PSGs [50]. Considering the non-essentiality of the yeast CSN, the above may lead to a drift of Rpn5 toward the essential proteasome, hence resulting in CSN malfunction, which could explain the accumulation of yCul1^R^. Indeed, impairment of proteasome lid assembly was shown to be associated with aggregation of proteasome subunits in subcellular stress foci as insoluble aggregates known as insoluble protein deposit (IPOD) and with aggregation of ubiquitinated unfolded proteins in a site of sequestration, known as juxtanuclear quality control (JUNQ) [48,60]. To evaluate the possible aggregation of Rpn5, we examined the distribution of a genomic Rpn5-GFP in *rpn11-m1* mutant strain, transformed independently with a plasmid of either Gal1p-mCherry-VHL (marker for IPOD) or Gal1p-mCherry-Rnq1 (marker for JUNQ). Cells were grown to the post-diauxic phase at 28 °C and at 34 °C (to induce aggregation), and co-localization was assessed using a confocal microscope (Figure 3).

Consistent with previous studies and in comparison to WT, the localization of Rpn5-GFP in *rpn11-m1* at 28 °C was primarily nuclear with some cytosolic dispersion [50]. Rpn5-GFP was mostly soluble in the mutant, except for a slight co-localization with JUNQ (Figure 3A, bottom). At 34 °C, unlike WT, the distribution of Rpn5-GFP in *rpn11-m1* was mainly cytosolic [50,61], with minor co-localization with IPOD and JUNQ (Figure 3A; right). This confirms that Rpn5-GFP in *rpn11-m1* is soluble, even though it is partially improperly co-localized with the CSN that is predominantly nuclear in budding yeast [26]. To examine if high yCul1^R^ status is due to deprivation of Rpn5, yCul1 NEDDylation status was assessed in *rpn11-m1* upon ectopic expression of *RPN5* through plasmids that suppress CSN structure or function deficiency of *rpn5-1* (a carboxyl terminal truncated mutant of Rpn5 that cannot integrate into the CSN complex) [30]. However, the overexpression of Rpn5 did not alter the yCul1^R^ to yCul1 ratio in *rpn11-m1* (Figure 3B). Taken together, *rpn11-m1* includes sufficient levels of Rpn5, and hence a direct involvement of Rpn11 enzymatic activity in determining the yCul1^R^ to yCul1 ratio is proposed. 

### 3.4. Rpn11-Mediated deubiquitinase Activity Authorizes COP9 signalosome-Mediated deNEDDylation of yCul1 

Since cullin deNEDDylation is not an inherent enzymatic property of the proteasome [30], Rpn11 DUB activity might regulate the hydrolysis of Rub1 from yCul1 by another mechanism. Hydrolysis of Ub by intact 19S-lid harboring active Rpn11 is coupled with translocation of substrates into the 19S ATPases ring for unfolding, therefore activity of this DUB is ATP-dependent [2,62]. Previous studies suggested a direct interaction between the 19S proteasome and the SCF upon ATP depletion, causing the stabilization of classic SCF substrates and suggesting their improper translocation [62]. We hypothesized that Rpn11 dysfunction would lead to stabilization of these 19S proteasome–SCF interactions. To evaluate this hypothesis, we examined SCF–proteasome interactions in WT and *rpn11-m1* mutant cells by reciprocal SCF and proteasome affinity purifications (AP) in the absence of ATP (Figure 4A,B). We used only *rpn11-m1* and not *rpn11^D122/A^* for this and other experiments, as expression of plasmids and harsh growth conditions led to loss of vitality of the latter. In agreement with earlier comparative mass spectrometry analysis, calmodulin-based purification of TAP-Rpt6 proteasomes revealed a loss of Rpn11 alongside other 19S-lid subunits in *rpn11-m1* isolates (Figure 4A) [43]. Nevertheless, similar to WT, TAP-Rpt6 proteasomes of *rpn11-m1* co-precipitated with yCul1 and even pulled out together with Cdc4, suggesting an interaction between the 19S proteasome and an assembled SCF^Cdc4^ complex (Figure 4A,B). Reciprocal affinity purification of SCF through RGS-8His-yCul1 was pulled out together with HMW Ub conjugates in *rpn11-m1* but not in WT. The co-purification of ubiquitinated proteins with RGS-8His-yCul1 represent CRL-ubiquitinated substrates. Unlike the results obtained with *rpn11-m1*, enrichment of high molecular weight Ub chains in total extracts of WT cells pretreated by the proteasome inhibitor MG132 did not elute with the SCF (Figure 4B). The data in Figure 4 suggest that the observed effects are unlikely to stem from depletion of free Ub, as although the treatment with MG132 leads to mono-Ub depletion, it does not affect the RGS-8His-yCul1–Ub interaction. Our data show that upon Rpn11 malfunction, the 19S proteasome interacts with assembled SCF complexes loaded with ubiquitinated substrates.

Considering that CSN activity occurs only after the dissociation of targets from the SCF [28,63,64,65], the accumulation of yCul1^R^ in *rpn11* mutants at the post-diauxic phase could be explained by the tight attraction between the ubiquitinated targets and the SCF (Figure 4A,B), which prevents CSN accessibility. To assess this hypothesis, we first confirmed that the CSN is active at the post-diauxic phase by treating *Δpdr5* cells lacking the ABC drug transporter Pdr5 with a specific inhibitor of Csn5, CSN5i-3 [45], and calculated the ratio of yCul1^R^ in CSN5i-3 treated and untreated cells (Figure 4C; 24 h in grey). Nevertheless, unlike the effect on *Δpdr5* cells, the inhibitor did not have any effect on *Δpdr5* cells also carrying the *rpn11-m1* mutation, suggesting absence of Cullin deNEDDylation in this mutant (Figure 4C, black). The above proposes DUB activity of the 19S-lid as a prerequisite for vacating the SCF and enabling CSN activity, hence adding Rpn11 to the SCF pathway.

A hallmark of *csn* deletion mutants and of *Δrub1* is synthetic sickness or lethality with various SCF mutants (i.e., *cdc53-1, skp1-12, cdc4-1,* and *cdc34-2* [25,66]). Similarly, decreased viability of *rpn11-m1–Δcsn* or *rpn11-m1–Δrub1* double mutants in W303 and BY4741 laboratory strains has been measured (Figure 5A and Appendix A). Furthermore, microscopic examination and DAPI staining of double mutants revealed swollen cells with cytokinesis defects, including diffused *mt*DNA morphology, suggesting additional mitochondrial defects (Figure 5B). Notably, due to the high rate of reversion in *rpn11-m1*, viability of double mutants was assessed at the first generation of growth from spores instead of the commonly used drop assay (Figure 5A,B and Appendix A) [67].

## 4. Conclusions

We described two distinct *rpn11* mutants that uniquely accumulate Ub and Rub1 modifiers, covalently attached to their substrates and attracted to SCF complexes. Our data suggest that malfunctioned Rpn11 DUB activity inhibits Cullin deNEDDylation, probably by preventing the CSN from reaching its yCul1^R^ substrate through spatial interference of SCF complexes charged with ubiquitinated proteins and co-interacting with the proteasome. Notably, equilibrium bias of Rub1 and Ub in favor of conjugate formation attached to the SCF could also indicate increased activity of their enzymatic cascades, both of which depend on Rbx1 as an E3 ligase [68]. This hypothesis should be further investigated in the future. Our data on Rpn11 activity as a prerequisite for CSN activity is in agreement with previous studies showing that depletion of ATP inhibits the CSN-mediated deNEDDylation reaction rate [69], which possibly indicates inhibition of the 19S ATPases ring and translocation of substrates. Direct SCF–proteasome interactions have been reported previously [62], raising the possibility that SCF unloads ubiquitinated cargo directly to the 26S proteasome for proteolysis. It is still unclear what mediates interaction of SCF complexes with proteasomes and whether the interaction is direct. Evidently, this interaction is not through subunits lacking in *rpn11-m1* proteasomes (Figure 4B [43]), nor through the poly-Ub chain or Rub1 [62]. Clearly, the significance of this interaction is that the proteasome acts to determine the recycling and reactivation of upstream CRL-ubiquitination enzymes. In summary, by studying the accumulation of yCul1^R^ conjugates in *rpn11* mutants we were able to distinguish between two molecular mechanisms that contribute to Cullin NEDDylation cycles: the previously studied redox homeostasis [41] and proteasome capacity (the current study). Apparently, a reducing environment supports Cullin NEDDylation, whereas proper proteasome function promotes Cullin deNEDDylation (Figure 6).

## Figures and Tables

**Figure 1 biomolecules-09-00449-f001:**
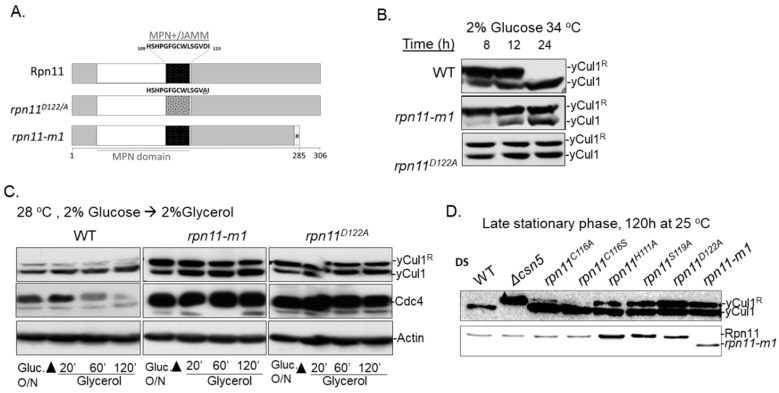
Distinct *rpn11* mutants exhibit high yCul1 NEDDylation status at the post diauxic phase. (**A**) Schematic representation of WT (wild type) Rpn11 and the distinct mutants of *rpn11-m1* and *rpn11^D122/A^*. WT Rpn11 includes 306 amino acids and bears an MPN domain at the amino terminal domain (white) with an MPN+–JAMM motif (black). This motif is mutated in *rpn11^D122/A^*, leading to defects in metalloprotease activity (dotted square). The mutant of *rpn11-m1* is shorter due to a frameshift, causing the absence of 31 carboxyl terminal amino acids and their replacement by nine other amino acids. (**B**) WT and mutated Rpn11 (*rpn11-m1* and *rpn11^D122/A^*) cells at the post-diauxic phase were diluted in YPD (yeast extract, peptone, dextrose) to 0.5 OD_600_ and grown at 34 °C. The modification status of yCul1 was examined by immunoblotting at indicated time points. Note: DS—diauxic shift. (**C**) WT and *rpn11* mutants (*rpn11-m1* and *rpn11^D122/A^*) at the post-diauxic phase were pre-washed and diluted to 0.5OD_600_ in YPG (yeast extract, peptone, glycerol). The modification status of yCul1 was examined by immunoblotting at indicated time points. (**D**) Overnight-grown WT, *Δcsn5*, and various mutants of Rpn11 with a silent mutation (*rpn11^C116A^*, *rpn11^C116S^*) or active site dead mutation (*rpn11^S119A^*, *rpn11^D122A^*) were diluted to 0.5OD_600_ and grown in YPD at 25 °C for 120 h before examination of yCul1 modification status by immunoblotting. Notably, a permissive temperature of 25 °C was used for the long-term growth (120 h) of the *rpn11^D122/A^* mutant to prevent growth defects and lethality [37].

**Figure 2 biomolecules-09-00449-f002:**
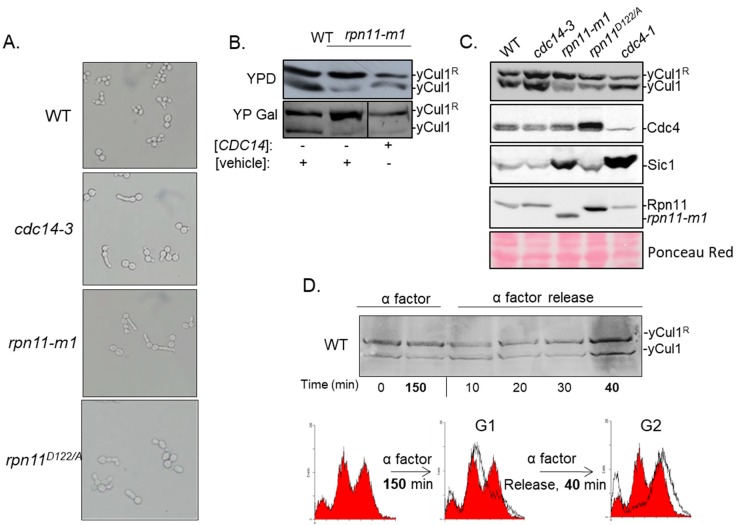
Cell cycle does not determine yCul1 NEDDylation status. (**A**) WT and mutant yeast strains were grown at 28 °C for 16 h, then shifted to a restrictive temperature of 37 °C for 5 h to enhance cellular phenotypes. (**A**) Cell cycle defects were observed by light microscope (magnification x400). (**B**) The *rpn11-m1* mutant cells were grown overnight in raffinose and diluted to 0.5 OD_600_ in galactose (YP Gal) to induce the expression of *CDC14* or a vehicle plasmid, or in glucose (YPD) for control. The extent of yCul1 modification by Rub1 was examined by immunoblotting of total protein extracts with yCul1 antibody. (**C**) Yeast strains were grown as previously explained (**A**), followed by extraction of total protein extracts used for yCul1 immunoblotting. The accumulation of the short-lived F-box protein Cdc4 and its substrate Sic1 was validated as well. Note that *cdc4-1* is a control for decreased endogenous levels of the yCul1^Cdc4^ complex. Cdc4 expression shows a negative correlation with the accumulated Sic1. (**D**) WT cells were grown in glucose supplemented by α factor for 150 min for synchronization of the cell cycle in G1. Synchronized cells were washed and were then grown in YPD. Samples were taken before and after synchronization at indicated times and subjected to immunoblotting with yCul1. The cell cycle stage was analyzed by quantitation of DNA content by flow cytometry.

**Figure 3 biomolecules-09-00449-f003:**
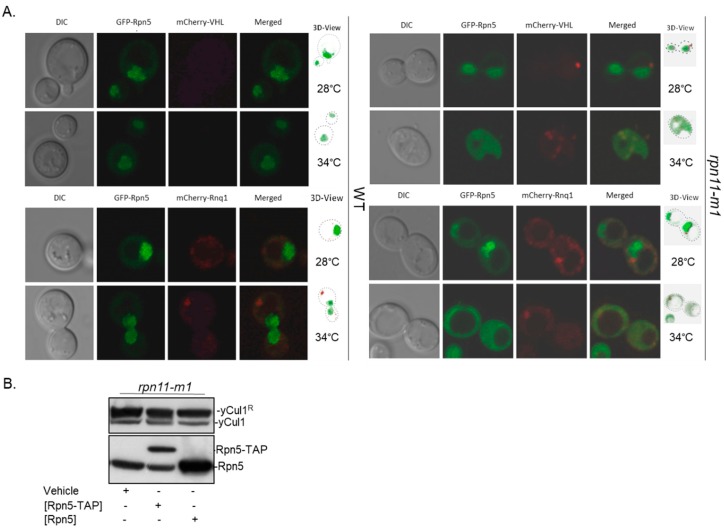
Intracellular distribution of Rpn5 in *rpn11-m1*. (**A**) WT (left) and *rpn11-m1* mutant cells (right) expressing a genomic copy of Rpn5–GFP and plasmids expressing either mCherry-VHL (top) or mCherry-Rnq1 (bottom). Cells were grown at indicated temperatures for 5 h and images of living cells were taken by ×100 oil immersion objective using a confocal microscope (Zeiss LSM 510). Z stacked images were captured at 0.5 μM intervals and images were processed using Zen Lite software. (**B**) Total cell extracts from *rpn11-m1* constitutively expressing Rpn5-TAP under ADH1 promoter or expressing Rpn5 under the control of *RPN10* promoter were resolved in polyacrylamide gel and blotted with anti-yCul1 and anti-Rpn5.

**Figure 4 biomolecules-09-00449-f004:**
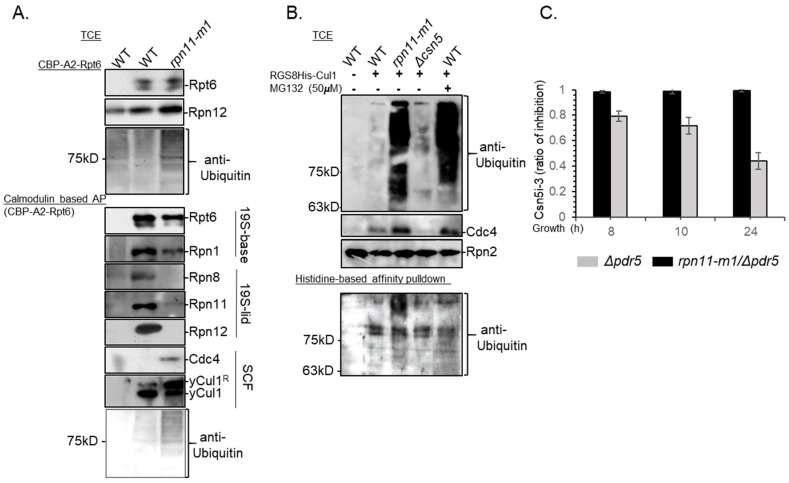
SCF complexes of *rpn11-m1* are loaded with ubiquitinated substrates and co-interact with the 19S proteasome. (**A**) WT and *rpn11-m1* mutant cells were transformed with a plasmid encoding to Rpt6, amino-terminally tagged by a calmodulin binding peptide and two repeats of protein A (CBP-A2-Rpt6). Total cell extracts (TCE) of overnight-grown cells were subjected to calmodulin-based affinity purification (AP). Co-purification of TAP-Rpt6 with other 19S proteasome subunits (Rpn1, Rpn8, Rpn11, Rpn12), SCF subunits (yCul1, Cdc4), and Ub was examined by immunoblotting. (**B**) WT, *rpn11-m1*, and *Δcsn5* were transformed with a plasmid expressing RGS-8HIS-yCul1. Cells were grown for 24 h at 28 °C followed by a treatment of WT culture with 50 µM of the proteasome inhibitor MG132 for 4 h. Total cell extracts (TCE) were used for a Histidine-based affinity pulldown of the RGS-8HIS-yCul1 with co-purified proteins. Defects in proteasome function (in *rpn11-m1* and WT treated by MG132) were evaluated by the accumulation of the short-lived protein Cdc4 and Ub conjugates (top and middle). Native cell extracts were used for Ni-NTA (nickel-charged affinity resin) based pulldown of RGS-8HIS-yCul1, and co-purification with Ub conjugates was examined by immunoblotting (bottom). (**C**) Overnight diluted culture were grown for an additional 6 h before the addition of 20 µM CSN5i-3 to the cells. The accumulation of yCul1^R^ in treated and untreated cells was evaluated by immunoblotting in three repeats, followed by quantification of densitometry by IMAGEJ, and calculation of the ratio between treated and untreated cells at indicated times (8, 10, or 24 h).

**Figure 5 biomolecules-09-00449-f005:**
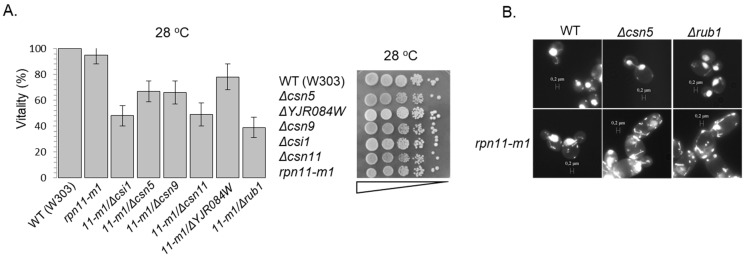
Rpn11 is a component of the SCF axis. (**A**) Vitality of double mutants cells was determined by the loss of their ability to from colonies upon plating on glucose rich agar plates at a permissive temperature. Experiments were done thrice (*n* = 3) and statistical significance was confirmed (left). Vitality of single mutants was determined through a drop assay by plating cultures in serial dilution in YPD agar dishes (right). Appendix A includes complementary data. (**B**) Tetrads of WT, single or double mutant strains, or mutant yeast strains separated into spores and the annotated strains grown to the first logarithmic phase in the permissive temperature of 28 °C. Cells were stained for nucleic acids by DAPI (4′,6-diamidino-2-phenylindole) and observed by ×63 oil immersion objective using a florescence microscope. In the double mutants, an aberrant cell phenotype reminiscent of the semi-lethal phenotype can be observed.

**Figure 6 biomolecules-09-00449-f006:**
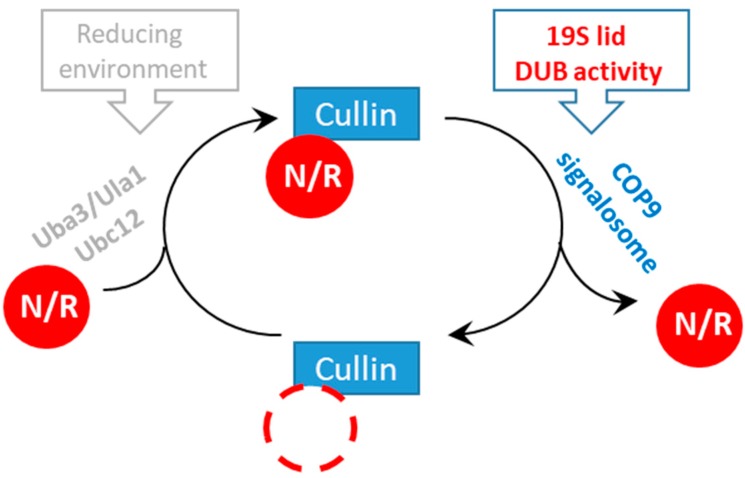
Regulation of Cullin modification cycles by the proteasome. Mutants of the 19S-lid subunit Rpn11 lead to the finding that catalytically active Rpn11 in proteasomes authorizes Cullin deNEDDylation by the CSN.

**Table 1 biomolecules-09-00449-t001:** Plasmids used in this study.

	Name	Description	Source
EP25	Csn5-TAP	*GAL1p* [CSN5-TAP]	Open Biosystems
EP53	empty vector	Yeplac181	
EP134	CDC14-GFP	*GAL1p* [CDC14-GFP], Amp	[46]
EP149	empty vector	pYes2	
EP150	pYC-RPN8	*pADH1*, [RPN8], Amp	[47]
EP228	mch-VHL	*GAL1p* [VHL-mCherry], Amp	[48]
EP229	mch-Rnq1	*GAL1p* [Rnq1-mCherry], Amp	[48]
EP204	CBP -Rpt6	*RPT4p*, [CBP-A2-RPT6-LEU2], Amp	[30]
EP234	Rpn5-TAP	*ADH1p* [Rpn5-TAP]	Open Biosystems
EP235	ScRpn5	*RPN10p* [RPN5], Amp	[30]
M134	RPN11 C116>A	YCPlac111, *RPN11p* [*rpn11^C116/A^*], Amp	[37]
M138	RPN11 D116>S	YCPlac111, *RPN11p* [*rpn11^C116/S^*], Amp	[37]
M143	RPN11 H111>A	YCPlac111, *RPN11p* [*rpn11^H111/A^*], Amp	[37]
M144	RPN11 S119>A	YCPlac111, *RPN11p* [*rpn11^S119/A^*], Amp	[37]
M145	RPN11 D122>A	YCPlac111, *RPN11p* [*rpn11^H122/A^*], Amp	[37]

**Table 2 biomolecules-09-00449-t002:** *Saccharomyces cerevisiae* strains used in this study.

Name	Strain	Genotype	Source
RC1	*Δcsn9*	*W303: csn9:: G418*	This study
RC6	*rpn11-m1–Δcsi1*	*W303**ade2-1; can1-100; his3-11,15; leu2-3, trp1-1; ura3-1; GAL+; lys2**, KanMX4**::YMR025W*	This study
RC13	*rpn11-m1–Δcsn9*	*W303 ade2-1; can1-100; his3-11, 15; leu2-3, trp1-1; ura3-1; GAL+; lys2, KanMX4::YDR079C*	This study
RC21	*ΔYJR084W*	*W303 YJR084W:: G418*	This study
RC22	*rpn11-m1–ΔYJR084W*	*W303 ade2-1; can1-100; his3-11,15; leu2-3, trp1-1; ura3-1; GAL+; lys2::ΔYJR084W*	This study
RC25	*Δcsi1*	*W303: csi1:: G418*	This study
YP61	*∆ubp6*	BY4741 *lys2-801 leu2-3, 2-112, ura3-52, his3-Δ200, trp1-1, Δubp6::HIS3*	Open Biosystems
YP76	*∆ubp15*	BY4741 *his3Δ1 leu2Δ0 ura3Δ0 met15Δ1, UBP::KanMX*4	Open Biosystems
YP77	*∆ubp16*	BY4741; *his3Δ1 leu2Δ0 ura3Δ0 met15Δ1, UBP::KanMX*4	Open Biosystems
YP86	*∆ubp1*	BY4741 *his3Δ1 leu2Δ0 ura3Δ0 met15Δ1, UBP::KanMX*	EUROSCARF (Oberursel, Germany)
YP87	*∆ubp2*	BY4741 *his3D1; leu2D0; met15D0; ura3D0; YOR124c::kanMX4*	EUROSCARF (Oberursel, Germany)
YP89	*∆ubp5*	BY4741 *his3Δ1 leu2Δ0 ura3Δ0 met15Δ1, UBP::KanMX*4	Open Biosystems
YP90	*∆ubp7*	BY4741 *his3Δ1 leu2Δ0 ura3Δ0 met15Δ1, UBP::KanMX4*	Open Biosystems
YP91	*∆ubp8*	BY4741 *his3Δ1 leu2Δ0 ura3Δ0 met15Δ1, UBP::KanMX*4	Open Biosystems
YP92	*∆ubp9*	BY4741 *his3Δ1 leu2Δ0 ura3Δ0 met15Δ1, UBP::KanMX*4	Open Biosystems
YP94	*∆ubp11*	BY4741 *his3Δ1 leu2Δ0 ura3Δ0 met15Δ1, UBP::KanMX*4	Open Biosystems
YP207	*Δcsn11*	*W303: csn11:: G418*	This study
YP212	*rpn11-m1–Δcsn5*	*W303**ade2-1; can1-100; his3-11,15; leu2-3, trp1-1; ura3-1; GAL+; lys2**, KanMX4**::YDL216C*	This study
YP216	*rpn11-m1–Δcsn11*	*W303* ade2-1; can1-100; his3-11,15; leu2-3, trp1-1; ura3-1; GAL+; lys2, *KanMX4::YIL071C*	This study
YP334	*W303 (parental)*	*Mat a, his3-200, ade2-101, leu21,ura3-52, lys2-801, trp162*	
YP335	*Δcsn5*	W303: *Csn5:: KanMX4*	[49]
YP336	*Δrub1*	W303 *his3ko1;leu2ko0;met15ko0;ura3ko0* *YDR139c::kanMX4*	This study
YP337	*rpn11-m1*	W303 *Mat a, his3_-200, ade2-101, leu2_1, ura3-52, lys2-801,trp1-62, YFR004W::rpn11-m1*	[47]
YP238	*rpn11-m1–Δcsn5*	W303 *Mat a, his3_-200, ade2-101, leu2_1, ura3-52, lys2-801,trp1-62, YFR004W::rpn11-m1, Csn5:: KanMX4*	This study
YP339	*rpn11-m1–Δrub1*	W303 *Mat a, his3_-200, ade2-101, leu2_1, ura3-52, lys2-801,trp1-62, YFR004W::rpn11-m1*, *YDR139c::kanMX4*	This study
YP444	*Rpn11-Rpn5-GFP*	*W303 MATa leu2-3,112 trp1-1 can1-100 ura3-1 ade2-1 his3-11,15 RPN11:3HA-KANMX6, RPN5:GFP(S65T)-TRP1*	[50]
YP445	*rpn11-m1-Rpn5-GFP*	*W303 MATa leu2-3,112 trp1-1 can1-100 ura3-1 ade2-1 his3-11,15 rpn11-m1:3HA-KANMX6, RPN5:GFP(S65T)-TRP1*	[50]
YP452	*cdc14-3*	BY4741 *his3ko1;leu2ko0;lys2ko0;ura3ko0*	[46]
YP531	*cdc4-1*	*W303 ura3-1,can1-100, gal+, leu2-3,112. trp1-1; ade2-1; his3-11,15; cdc4-1*	[51]
MY321	*rpn11^D122/A^*	BY4741 *his3ko1; leu2ko0; met15ko0; ura3ko0 YFR004W:kanMX4 with plasmid M145*	[37]
MY317	*rpn11^C116/A^*	BY4741 *his3ko1; leu2ko0; met15ko0; ura3ko0 YFR004W:kanMX4 with plasmid M134*	[37]
MY318	*rpn11^C116/S^*	BY4741 *his3ko1; leu2ko0; met15ko0; ura3ko0 YFR004W:kanMX4 with plasmid M138*	[37]
MY319	*rpn11^H111/A^*	BY4741 *his3ko1; leu2ko0; met15ko0; ura3ko0 YFR004W:kanMX4 with plasmid M143*	[37]
MY320	*rpn11^S119/A^*	BY4741 *his3ko1; leu2ko0; met15ko0; ura3ko0 YFR004W:kanMX4 with plasmid M144*	[37]
MY1021	*Δpdr5*	W303 *ura3-1; can1-100; GAL+leu3,112,trp1-1; ade2-1; his3-11,15; pdr5::hisG*	[52]
MY1424	*rpn11-m1–Δpdr5*	W303 *ura3-1; can1-100; GAL+leu3,112,trp1-1; ade2-1; his3-11,15; pdr5::hisG*; *YFR004W::rpn11-m1*	This study

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
