# Peer review of "The Proteasome Lid Triggers COP9 Signalosome Activity during the Transition of Saccharomyces cerevisiae Cells into Quiescence"

_biomolecules, 2019, doi:10.3390/biom9090449_

Round 1
Reviewer 1 Report
In this article, Bramasole et al make an attempt to study the cause of increased Neddylation of Cullin 1 (Cul1) when Rpn11 mutants (a catalytic inactive version and a deletion of the last 34 amino acids) are expressed in S. cerevisiae. Rpn11 is a de-ubiquitinating enzyme associated with the proteasome. The authors observed that a mutant in Rpn11 (Rpn11-m1) produces a change in the distribution of Rpn5, a component of the Rpn5 19S-lid (and also a component of the signalosome) and they link that with the Cul1-modified form.
The manuscript contains some interesting data but needs to be improved before publication.
Major comments:
1- Figure 1C: The authors say that “the high ratio of yCul1R in rpn11 mutants correlates with accumulation of the short‐lived Fbp, Cdc4, probably due to defects in proteolysis”. This should be confirmed by measuring the half-life (for example by adding cycloheximide) of Cdc4 in the rpn11-expressing mutants, compared to WT strains. In addition, studying the protein levels and/or half-life of other proteins that are degraded by Cul1 will add interest in the article.
2- In Figure S2B an interesting study is performed with other DUBs to address the nature of the yCul1R in rpn11 mutant strains. However, the results shown are not very clear (for example the ratio of the 2 yCul1 bands in the WT seem to be different than in other experiment). It would be better to repeat this experiment in YP-Glycerol for 72 hours (when there is no Cul1R band in the WT).
3- In Figure 2B the authors claim that a correlation exists between yCul1R in rpn11 mutants higher levels of Sic1 l. However, I do not see that correlation with the D122A mutant (it is clear for the rp11-m1 mutant). To verify the data, a loading control shouId be included and the differences should be quantified.
4- I cannot see the yCul1 unmodified in the WT panel in Figure 2C. This blot should be repeated. In addition, the expression of CDC14 should be studied.
5- Figure 2D shows no changes in yCul1R during the cell cycle in a WT strain. This experiment should also be performed in the rpn11-m1 strain.
6- An essential control in Figure 3 would be to check GFP-Rpn5 localization in a WT strain at 28 and 34C.
7- The authors claim that the most important finding of the article is the one of figure 3, but I miss a link to the yCul1R modified form. Could a Rpn5 mutant (or similar) be used to study the yCul1 status?. This would increase the impact of their finding.
8- What exactly is analysed in the Western blot in the lower panel of Fig 4B? Is it against Cul1?
9- I suggest that that authors try to study if a direct interaction between yCul1 and Rpn11 (WT and mutants) exists. It is an obvious experiment, but not shown.
9- In Figure 5A many controls of viability in single mutants (Dcsi1, DCsn11, DYR084W and Drub1) are missing. These should be added.
Minor comments:
1- Lines 20-21 in the abstract: half a sentence is duplicated.
2- What is “DS” in Figure 1B?
3- In line 254-355, I would include the name mCherry instead of “mch” to avoid confusions.
4- Figure 4A: “Calmodulin based AP”. What is AP, “affinity purification”? Then state the abbreviation.
5- I strongly suggest making all figures coherent: for example in the top of the western blots, sometimes the labelling is at +45 degrees and sometimes at -45 degrees.
Author Response
Reviewer #1:
Major comments:
1- Figure 1C: The authors say that “the high ratio of yCul1R in rpn11 mutants correlates with accumulation of the short‐lived Fbp, Cdc4, probably due to defects in proteolysis”. This should be confirmed by measuring the half-life (for example by adding cycloheximide) of Cdc4 in the rpn11-expressing mutants, compared to WT strains. In addition, studying the protein levels and/or half-life of other proteins that are degraded by Cul1 will add interest in the article.
Cdc4 is a well-known short-lived protein, self-ubiquitinated by the SCF and degraded by the proteasome. Hence, we used the accumulation of Cdc4 as a marker for proteasome inhibition in both mutants. Nevertheless, according to this suggestion, we have now added to a new cycloheximide (CHX) chase experiment (Figure S1C), wherein the decay in the steady-state level of Cdc4 was monitored in WT and rpn11-m1 by immunoblotting.
2- In Figure S2B an interesting study is performed with other DUBs to address the nature of the yCul1R in rpn11 mutant strains. However, the results shown are not very clear (for example, the ratio of the 2 yCul1 bands in the WT seem to be different than in other experiment). It would be better to repeat this experiment in YP-Glycerol for 72 hours (when there is no Cul1R band in the WT).
We thank the reviewer for the comment. According to the suggestion, we repeated this experiment and validated yCul1 NEDDylation status in the deletion mutant strains of 14 DUBs. As recommended by the reviewer, yCul1 NEDDylation and the expression of Rpn11 were evaluated at the post-diauxic phase in both glucose- and glycerol- YP medium (Figure S3).
3- In Figure 2B the authors claim that a correlation exists between yCul1R in rpn11 mutants higher levels of Sic1 l. However, I do not see that correlation with the D122A mutant (it is clear for the rp11-m1 mutant).
We thank the reviewer for arising this point. Our results show accumulation of Sic1 in both of rpn11 mutants; however, the accumulation in rpn11-m1 is consistently stronger than in the D122/A mutant. On the other hand, Cdc4 (an F box protein and also an SCF substrate) is highly accumulated specifically in the D122/A mutant. We assume that the difference in expression of these proteins is related with unique properties of each mutant. For example, the accumulation of Sic1 (by both increased synthesis and decreased degradation) depends on Cdc14 phosphatase (Visintin, R. et al. 1998 Mol. Cell); and Cdc14 phosphatase activity depends on the CTD of Rpn11 that is missing in rpn11-m1 but existing in the D122/A mutant (Esposito, M.et al. 2011). We added description of Cdc4 and Cdc14 to the introduction and mentioned this issue at the results part (lines 44-52). We also added an immunoblot for Rpn11 and ponceau red staining to confirm the truncation in rpn11-m1 and the equal loading of total proteins (Figure 2C), and described the results (lines 239-242).
4- I cannot see the yCul1 unmodified in the WT panel in Figure 2C. This blot should be repeated. In addition, the expression of CDC14 should be studied.
We cropped the band of WT from the figure since the expression of yCul1 in WT is not relevant here and +/- over expression of Cdc14 is a sufficient control.
5- Figure 2D shows no changes in yCul1R during the cell cycle in a WT strain. This experiment should also be performed in the rpn11-m1 strain.
We respectfully disagree with this comment. The treatment of WT with alpha factor was performed to arrest cell-cycle and mimic the physiological phenotype of Rpn11 mutants. The experiment cannot be performed in rpn11 mutants since (i.) the cell cycle of the mutants is already arrested (see the phenotype in figure 2A) and (ii.) these mutants show constitutive accumulation of yCul1R.
6- An essential control in Figure 3 would be to check GFP-Rpn5 localization in a WT strain at 28 and 34C.
This is a valuable suggestion and we have added them to the Figure.
7- The authors claim that the most important finding of the article is the one of figure 3, but I miss a link to the yCul1R modified form. Could a Rpn5 mutant (or similar) be used to study the yCul1 status? This would increase the impact of their finding.
We apologize for not being clear. We believe that the most important finding in this manuscript is within Figure 4 and not Figure 3. Figure 3 shows that Rpn5, even if overexpressed, does not affect cullin NEDDyation status. This is also explained it in the text (lines 296-299). In addition, yCul1 cannot be studied in rpn5 mutants since Rpn5 is a subunit of the COP9 signalosome, as was nicely described by this reviewer at the opening remarks, and also mentioned in the introduction (lines 73-74) and at the results (line 266). Therefore, rpn5 mutants constitutively show total accumulation of yCul1R (Yu et al. 2011).
8- What exactly is analysed in the Western blot in the lower panel of Fig 4B? Is it against Cul1?
The immunoblot is against ubiquitin. It is now corrected (from "Ubiquitin" to "anti-ubiquitin").
9- I suggest that that authors try to study if a direct interaction between yCul1 and Rpn11 (WT and mutants) exists. It is an obvious experiment, but not shown.
Our results show that assembled SCF co-interacts with the proteasome even without Rpn11 (Figure 4A, right lane), though, Rpn11 could not be the link between the proteasome and the SCF.
9- In Figure 5A many controls of viability in single mutants (Dcsi1, DCsn11, DYR084W and Drub1) are missing. These should be added.
Sporadic double mutants were prepared at the BY4741 background, only to confirm conservation of synthetic sickness in various yeast genotypes. According to this comment, we relocated the BY4741 results to the supplementary materials (S Figure 5).
Minor comments:
1- Lines 20-21 in the abstract: half a sentence is duplicated.
Corrected
2- What is “DS” in Figure 1B?
DS is diauxic shift, it is now described at fig' legends.
3- In line 254-355, I would include the name mCherry instead of “mch” to avoid confusions. Thank you!
We have replaced all mch with mCherry and added a description in the figure legend.
4- Figure 4A: “Calmodulin based AP”. What is AP, “affinity purification”? Then state the abbreviation. Abbreviation will be added to the legends.
Indeed, AP is affinity purification. We added the abbreviation in respective figure legend.
5- I strongly suggest making all figures coherent: for example in the top of the western blots, sometimes the labelling is at +45 degrees and sometimes at -45 degrees.
We thank the reviewer for this suggestion. We followed this suggestion and edited the figures accordingly.
Reviewer 2 Report
The manuscript by Bramasole et al. describes the identification of Rpn11 as a new regulatory component of the CRL function. Using S. cerevisiae as a model organism where the NEDD8/Rub1 pathway is not essential, the authors found that mutants of Rpn11, a component of the 26S proteasome lid, prevent the deNEDDylation of cullins during the post-diauxic shift, which is mediated by the action of the CSN complex. Further analysis indicates that the Rpn11 DUB activity within the proteasome complex is required for the deconjugation of NEDD8 from cullins by the CSN.
The study presents a striking observation on the functional link between the 26S proteasome and the CSN complex, an area of wide and long-term interest in the ubiquitin field. The data are convincing and well presented. I found the experiment in Fig. 4 particularly interesting, as it strongly indicates the role of Rpn11 and subsequently of the 26S proteasome in regulating CRL activity.
Points
1. In many (but not all) experiments, for example Fig 1D, it appears that Rpn11 mutants cause a general increase in cullin levels including the NEDDylated state. Do the authors have a comment on this? Does Rpn11 control the stability of cullins?
2. In Fig. 4, does the increase in the binding of ubiquitinated proteins to cul1 represent CRL ubiquitinated substrates? In other words does this ubiquitination depend on Rbx1 protein? Rbx1 may be interesting to test as it promotes both the cullin NEDDylation and substrate ubiquitination. In theory the phenotype of Rpn11 mutants could be due to hyperactivation of Rbx1?
3. I feel that many readers will raise the question of the role of the depletion of free ubiquitin under conditions of dysfunction of the proteasome activity in the cul1-ubiquitin binding assay. Based on the data in Fig. 4 where MG132, which causes ubiquitin depletion, but does not affect the cul1-ubiquitin interaction, the authors could comment that the observed effects are not likely due to depletion of free ubiquitin.
4. In Fig. 5 the authors show that the double Rpn11/CSN mutation reduces vitality, similarly to the double CSN/Rub1 mutant. However, based on the idea that Rpn11 and CSN target the same molecular event (hyperNEDDylation of cullins) it is not clear why the combination of these mutations should cause a synergism, especially when the CSN inhibitors do not affect CSN activity in Rpn11 mutants (Fig. 4C).
Author Response
Reviewer #2:
"The study presents a striking observation on the functional link between the 26S proteasome and the CSN complex, an area of wide and long-term interest in the ubiquitin field. The data are convincing and well presented. I found the experiment in Fig. 4 particularly interesting, as it strongly indicates the role of Rpn11 and subsequently of the 26S proteasome in regulating CRL activity".
We thank the reviewer for this statement.
Points
In many (but not all) experiments, for example Fig 1D, it appears that Rpn11 mutants cause a general increase in cullin levels including the NEDDylated state. Do the authors have a comment on this? Does Rpn11 control the stability of cullins?
We appreciate the reviewer for this observation. yCul1 is indeed a proteasome substrate. We have recently submitted a parallel paper explaining this observation.
In Fig. 4, does the increase in the binding of ubiquitinated proteins to cul1 represent CRL ubiquitinated substrates?
Yes. Figure 4 is entitled: "SCF complexes of rpn11-m1 are loaded with ubiquitinated substrates and co-interact with the 19S proteasome".
In other words, does this ubiquitination depend on Rbx1 protein? Rbx1 may be interesting to test as it promotes both the cullin NEDDylation and substrate ubiquitination. In theory the phenotype of Rpn11 mutants could be due to hyperactivation of Rbx1?
Regarding to Rbx1, we cannot exclude that hyperactivation of Rbx1 could lead to the same phenotypes (i.e. hyper-ub and hyper-NEDDylation). However, this option if occurs will not give an explanation for the decreased CSN activity (figure 4C) and the delayed release of ubiquitinated substrates from the E3 ligase complex. However, this is an interesting point. We will further investigate whether the phenotype is an outcome of a combined effect. According to this comment we added a sentence to the discussion (lines 272-274): "Notably, equilibrium bias of Rub1 and Ub in favor of conjugates formation attached to the SCF could also indicate on increased activity of their enzymatic cascades, both depend on Rbx1 as an E3 ligase".
I feel that many readers will raise the question of the role of the depletion of free ubiquitin under conditions of dysfunction of the proteasome activity in the cul1-ubiquitin binding assay. Based on the data in Fig. 4 where MG132, which causes ubiquitin depletion, but does not affect the cul1-ubiquitin interaction, the authors could comment that the observed effects are not likely due to depletion of free ubiquitin.
We thank the reviewer very much for this comment that is now incorporated in the text (lines 320-324): "Unlike the results obtained with rpn11-m1, enrichment of HMW Ub chains in total extracts of WT cells pretreated by the proteasome inhibitor MG132 did not elute with the SCF (Figure 4B). The data in figure 4 suggest that the observed effects are unlikely to stem from depletion of free Ub since the treatment with MG132, although leads to mono-Ub depletion, does not affect the RGS-8His-yCul1 - Ub interaction".
In Fig. 5 the authors show that the double Rpn11/CSN mutation reduces vitality, similarly to the double CSN/Rub1 mutant. However, based on the idea that Rpn11 and CSN target the same molecular event (hyperNEDDylation of cullins) it is not clear why the combination of these mutations should cause a synergism, especially when the CSN inhibitors do not affect CSN activity in Rpn11 mutants (Fig. 4C).
This is an interesting point. It is indeed anticipated that csn mutants will suppress defects of SCF (and Rpn11) mutants. Yet, paradoxically, multiple studies as well as high-throughput genetic screens described a negative genetic interaction between scf and csn/rub1 mutants. Accordingly, Cope et al. 2002 proposed an explanation to this anignma by suggesting that the CSN and Rub1 are needed to sustain optimal SCF activity. In this study we added Rpn11 to the SCF pathway by showing that in similar to scf mutants, the mutant of rpn11-m1 also shares a negative genetic interaction (determined by synthetic sickness) with csn/rub1 axis mutants. The relevant references are incorporated as well.
Reviewer 3 Report
The manuscript titled “Rpn11 DUB activity precedes cullin1 deNEDDylation activity of the COP9 signalosome during the transition of S. cerevisiae cells into quiescence" by Elah Pick, Lylane Bramasole, Abhishek Sinha, Angela Cirigliano, Gurevich Sylvia, Zanlin Yu, Dana Harshuk, Michael H. Glickman, Teresa Rinaldi is an important piece of work utilizing a yeast model system to investigate the molecular mechanism of Rub1-CRLs conjugation patters.
The manuscript founds itself on a well-summarized background with novel discoveries. However, the format for submission does not meet the satisfaction required for the Journal. To start with, the submitted title “Rpn11 DUB activity precedes cullin1 deNEDDylation activity of the COP9 signalosome during the transition of S. cerevisiae cells into quiescence” does not match the title on the manuscript “Proteasome lid triggers COP9 signalosome activity during the transition of S. cerevisiae cells into quiescence". The author order on the submission does not match the manuscript either “Elah Pick, Lylane Bramasole , Abhishek Sinha , Angela Cirigliano , Gurevich Sylvia , Zanlin Yu , Dana Harshuk , Michael H. Glickman , Teresa Rinaldi” vs “Lylan Bramasole, Abhishek Sinha, Angela Cirigliano, Sylvia Gurevich, Zanlin Yu, Dana Harshuk, Michael H. Glickman, Teresa Rinaldi, and Elah Pick”. Furthermore, some Figures do not match the text as noted in detail below. The manuscript should be carefully edited and read through before submission.
Having that said, the manuscript describes important findings. By utilizing Rpn11 mutants in S. cerevisiae, the manuscript reveals a proteasome lid-associated deubiquitination activity that authorizes the hydrolysis of Rub1 from yCul1 by CSN for the first time. Yet there are concerns in the manuscript that needs to be improved.
Typos: The text should be re-examined. For example, the following section should be rewritten.
1) L20-21: “upon entry into quiescence” is redundant in the sentence.
2) L65: “[24,28],,” has two commas.
3) L252: “sequestration(JUNQ)” should have a space in between.
Expression: Consider revising the expression to avoid misleading sentences.
1) L19: “abnormal accumulation of Rub1-modified cullin1 (yCul1)” should be made clear whether the accumulation of “Rub1-modified cullin1” was abnormal or the manner of modification of Rub1 was abnormal (such as the conjugation manner with substrates”.
2) L20, L41, L63: a comprehensive definition and consistent usage of “yCul1” “yCul1R” should be provided. It would be more comprehensive to the readers to use unified consistent expressions, especially for “yCul1R”.
L20: Rub1-modified cullin1 (yCul1)
L41: Cdc53/Cullin1 (herein yCul1) scaffold subunit
L63: cullin1‐Rub1 (yCul1R)
L74: modified Cullin‐1 (yCul1)
L77: hydrolysis of Rub1 from yCul1
L160: ratio of yCul1R to yCul1
3) Is it the asterisk in Fig1 equivalent to “yCul1R”, or is there a reason to note as a “representation” as noted in the legend?
4) Does DS in Figure 1B stand for diauxic shift? If so, should be noted in the legend. Furthermore the spelling for “diauxic shift” in the top of Supplementary Materials should be corrected.
5) L324: Figure 5 right panel “WT” should be made clear it is BY4741
6) L337: “Figure 5C” there is no “C” in the Figure.
Furthermore there are the following concerns.
The abstract sets a story to reveal “a negative feedback mechanism by proteasome capacity on upstream ubiquitination enzymes” (L23)” however the manuscript concludes with a stronger tone on “redox homeostasis”. It would be more comprehensive and less misleading if the message was consistent.
Author Response
Reviewer #3:
“The manuscript titled …... is an important piece of work utilizing a yeast model system to investigate the molecular mechanism of Rub1-CRLs conjugation patters”.
We thank the reviewer for this statement
“The author order on the submission does not match the manuscript either “Elah Pick, Lylane Bramasole , Abhishek Sinha , Angela Cirigliano , Gurevich Sylvia , Zanlin Yu , Dana Harshuk , Michael H. Glickman , Teresa Rinaldi” vs “Lylan Bramasole, Abhishek Sinha, Angela Cirigliano, Sylvia Gurevich, Zanlin Yu, Dana Harshuk, Michael H. Glickman, Teresa Rinaldi, and Elah Pick”.
We apologize about this mess. We experienced some difficulties upon uploading the manuscript. We contacted the editor (June 17 2019) with regards to this issue and received the following answer: “There is no problem about the abstract and title. What it is important is the file you correctly uploaded. Therefore, I confirm that it has been received correctly and it is already under review”.
Typos:
The text should be re-examined. For example, the following section should be rewritten.
L20-21: “upon entry into quiescence” is redundant in the sentence. Corrected L65: “[24,28]” has two commas. Corrected L252: “sequestration(JUNQ)” should have a space in between. Corrected
Expression:
Consider revising the expression to avoid misleading sentences:
1) L19: “abnormal accumulation of Rub1-modified cullin1 (yCul1)” should be made clear whether the accumulation of “Rub1-modified cullin1” was abnormal or the manner of modification of Rub1 was abnormal (such as the conjugation manner with substrates”.
The sentence is now corrected (lines 19-21): "In this study, we describe two distinct mutants of Rpn11, a proteasome-associated deubiquitinating enzyme, both of which exhibit a biochemical phenotype characterized by high accumulation of Rub1-modified Cdc53/cullin1 (yCul1) upon entry into quiescence in S. cerevisiae".
2) L20, L41, L63: a comprehensive definition and consistent usage of “yCul1” “yCul1R” should be provided. It would be more comprehensive to the readers to use unified consistent expressions, especially for “yCul1R”.
L20: Rub1-modified cullin1 (yCul1) L41: Cdc53/Cullin1 (herein yCul1) scaffold subunit L63: cullin1‐Rub1 (yCul1R) L74: modified Cullin‐1 (yCul1) L77: hydrolysis of Rub1 from yCul1
We thank the reviewer for this comment. We corrected this definition in the entire document.
3) L160: ratio of yCul1R to yCul13) Is it the asterisk in Fig1 equivalent to “yCul1R”, or is there a reason to note as a “representation” as noted in the legend?
Asterisks were replaced by "yCul1R" in all the figures
4) Does DS in Figure 1B stand for diauxic shift? If so, should be noted in the legend.
Corrected, see line 242 (under figure 1).
Furthermore, the spelling for “diauxic shift” in the top of Supplementary Materials should be corrected.
Thank you very much, it is now corrected.
5) L324: Figure 5 right panel “WT” should be made clear it is BY4741.
We added a description to this panel, which was removed to the supplementary materials (Figure S5).
6) L337: “Figure 5C” there is no “C” in the Figure. We apologize for this mistake.
Indeed.. We are sorry about it. it is now corrected.
The abstract sets a story to reveal “a negative feedback mechanism by proteasome capacity on upstream ubiquitination enzymes” (L23)” however the manuscript concludes with a stronger tone on “redox homeostasis”. It would be more comprehensive and less misleading if the message was consistent.
We deleted it from the abstract and diminished the description of REDOX in the scheme (Figure 6) and in the entire of the manuscript.
Round 2
Reviewer 1 Report
The manuscript improved but before publication authors need to improve the manuscript with the following:
1-Author:
Cdc4 is a well-known short-lived protein, self-ubiquitinated by the SCF and degraded by the proteasome. Hence, we used the accumulation of Cdc4 as a marker for proteasome inhibition in both mutants. Nevertheless, according to this suggestion, we have now added to a new cycloheximide (CHX) chase experiment (Figure S1C), wherein the decay in the steady-state level of Cdc4 was monitored in WT and rpn11-m1 by immunoblotting.
Reviewer- The new experiment in Figure S1C need to be quantified. T=0 equals to 1 (or 100) in each panel, as WT and rpn11-m1 are 2 different panels.
------
2-Reviewer V1: I cannot see the yCul1 unmodified in the WT panel in Figure 2C. This blot should be repeated. In addition, the expression of CDC14 should be studied.
Authors: We cropped the band of WT from the figure since the expression of yCul1 in WT is not relevant here and +/- over expression of Cdc14 is a sufficient control.
Reviewer V2- I do not think it is a correct and professional solution just to remove data because “is not relevant”. If the presented data are not good enough, do not include them in the first place. I insist that the authors repeat the blot.
In addition, in my opinion Cdc14 expression should be studied and shown.
----
3-Reviewer V1. In Figure 5A many controls of viability in single mutants (Dcsi1, DCsn11, DYR084W and Drub1) are missing. These should be added.
Authors: Sporadic double mutants were prepared at the BY4741 background, only to confirm conservation of synthetic sickness in various yeast genotypes. According to this comment, we relocated the BY4741 results to the supplementary materials (S Figure 5).
Reviewer V2- Authors do not show the experiment I suggested, just changed figures to supplemental data. I insist in new figure 5A, authors should measure vitality with W303 strains that are delta-csi1, delta-Csn11, delta-YR084W and delta-Drub1 only (not in combination with rpm11-m1), otherwise there is no meaning for the results shown.
Author Response
We thank the reviewer for highlighting the below points. We corrected the manuscript accordingly.
The new experiment in Figure S1C need to be quantified. T=0 equals to 1 (or 100) in each panel, as WT and rpn11-m1 are 2 different panels.
We have added a graph with average and standard deviation of 3 independent experiments (Figure S1C, bottom).
V2- I do not think it is a correct and professional solution just to remove data because “is not relevant”. If the presented data are not good enough, do not include them in the first place. I insist that the authors repeat the blot.
We incorporated a new blot instead of the previous one. The original IB was sent to the editor for confirmation.
In addition, in my opinion Cdc14 expression should be studied and shown.
Esposito M. et al. 2011 (DOI:10.1111/j.1567-1364.2010.00690.x) extensively studied the restoration of rpn11-m1 cell cycle phenotypes by this specific plasmid. Similarly, during our study, Cdc14 expression in rpn11-m1 had been validated by microscopy with every use of the plasmid. However, the photos are not representable. We sent them to the editor for confirmation and added a description to the text. See lines 240 – 242.
Reviewer V2- Authors do not show the experiment I suggested, just changed figures to supplemental data. I insist in new figure 5A, authors should measure vitality with W303 strains that are delta-csi1, delta-Csn11, delta-YR084W and delta-Drub1 only (not in combination with rpm11-m1), otherwise there is no meaning for the results shown
We apologize for not understanding the request at the first glance. The reviewer is definitely right, and the necessary controls have now been added. Since single CSN mutants are as viable as WT, we added a drop dilution assay, showing each of the single mutants (rpn11-m1, the delta csn mutants and delta YDR084W) to confirm this supposition (Figure 5A, right). A complementary description was added to the "methods part" (lines 110-114) and to the "figure legend".